# Foxn1-β5t transcriptional axis controls CD8+ T-cell production in the thymus

Muhammad Myn Uddin[1,*], Izumi Ohigashi[1,*], Ryo Motosugi[2,*], Tomomi Nakayama[2], Mie Sakata[1], Jun Hamazaki[2], Yasumasa Nishito[3], Immanuel Rode[4], Keiji Tanaka[5], Tatsuya Takemoto[6], Shigeo Murata[2] & Yousuke Takahama[1]

The thymus is an organ that produces functionally competent T cells that protect us from pathogens and malignancies. Foxn1 is a transcription factor that is essential for thymus organogenesis; however, the direct target for Foxn1 to actuate thymic T-cell production is unknown. Here we show that a Foxn1-binding cis-regulatory element promotes the transcription of β5t, which has an essential role in cortical thymic epithelial cells to induce positive selection of functionally competent CD8+ T cells. A point mutation in this genome element results in a defect in β5t expression and CD8+ T-cell production in mice. The results reveal a Foxn1-β5t transcriptional axis that governs CD8+ T-cell production in the thymus.

[1] Division of Experimental Immunology, Institute of Advanced Medical Sciences, University of Tokushima, Tokushima 770-8503, Japan. [2] Laboratory of Protein Metabolism, Graduate School of Pharmaceutical Sciences, University of Tokyo, Tokyo 113-0033, Japan. [3] Core Technology and Research Center, Tokyo Metropolitan Institute of Medical Science, Tokyo 156-8506, Japan. [4] Division of Cellular Immunology, German Cancer Research Center, D-69120 Heidelberg, Germany. [5] Laboratory of Protein Metabolism, Tokyo Metropolitan Institute of Medical Science, Tokyo 156-8506, Japan. [6] Laboratory for Embryology, Institute of Advanced Medical Sciences, University of Tokushima, Tokushima 770-8503, Japan. * These authors contributed equally to this work. Correspondence and requests for materials should be addressed to S.M. (email: smurata@mol.f.u-tokyo.ac.jp) or to Y.T. (email: takahama@genome.tokushima-u.ac.jp).

CD8[+] T cells have a central role in immune defense against viral infection, intracellular pathogens, and malignant tumours[1–3]. CD8[+] T cells are chiefly generated in the thymus through the process of positive selection[4,5]. Positive selection of functionally competent CD8[+] T cells is dependent on TCR engagement of immature thymocytes with self-peptides produced by the thymoproteasome, a thymus-specific form of the proteasome[6–10]. The thymoproteasome is specifically expressed in cortical thymic epithelial cells (cTECs) because its unique catalytic subunit β5t or Psmb11 is exclusively transcribed in cTECs[11–14]. However, how β5t is expressed specifically in cTECs is poorly understood. Previous studies showed that when the entire coding sequence of β5t in mouse genome is replaced with foreign sequences, including sequences encoding Venus fluorescence protein and Cre recombinase, the mouse retains cTEC-specific expression of Venus and Cre, respectively[6,12,13], suggesting that the genomic element that instructs the cTEC-specific expression of β5t is located mainly outside the β5t-coding sequence. Importantly, β5t expression in the embryonic thymus primordium is not detectable in Foxn1-deficient *nude* mice[11,14]. Foxn1 is a transcription factor that governs the development of TECs, and thymus organogenesis is prematurely arrested in Foxn1-deficient mice and humans[15,16]. However, whether Foxn1 directly controls the transcription of β5t or indirectly affects β5t by regulating molecules that are crucial for upstream TEC development has not been clarified.

Indeed, no direct targets of Foxn1 in its transcriptional regulation of gene expression have been identified in TECs, despite the importance of Foxn1 in TEC development. Like β5t, many molecules, including DLL4, CCL25 and PD-L1, expressed in TECs have markedly reduced expression in Foxn1-deficient mouse thymus[17–20]. However, it is not established whether Foxn1 directly or indirectly affects the expression of any of those TEC-associated genes, including functionally relevant genes in the thymus.

Here we report the identification of a highly conserved Foxn1-binding sequence that is located proximal to the β5t-coding sequence in the genome. *In vitro* experiments show that Foxn1 protein binds to this sequence and promotes proximal gene transcription. *In vivo* experiments in mouse show that this cis-regulatory element is indeed essential for the optimal expression of β5t in cTECs and the optimal production of CD8[+] T cells. Our results reveal a Foxn1-binding cis-regulatory element that is functionally relevant for the thymus to produce T cells.

## Results

### Foxn1-binding motifs adjacent to β5t-coding sequence.
β5t-encoding gene in the mouse genome is encoded by a single exon located within the 14-kb region between β5-encoding and Cdh24-encoding genes in chromosome 14 (ref. 6) (Fig. 1a). Within this 14-kb region, we searched for the 11-bp Foxn1-binding consensus motif, a a/g n g A C G C t a/t t, in which the middle tetranucleotide in large letters represented the invariant core motif[21]. Although there were no sites that perfectly matched this 11-bp consensus motif, we detected 18 sites that contained the 4-bp core motif (Fig. 1a,b). Among those 18 sites, site #13 located 80-bp upstream of *β5t* transcription initiation site best matched (2-bp mismatched) the 11-bp consensus motif (Fig. 1b). Among the four sites with second-best matched (3-bp mismatched) sequence, site #8 located 2.4-kb upstream of *β5t* transcription initiation site contained the longest (7-bp) region identical with the 11-bp consensus motif (Fig. 1b). The order and orientation of the neighbouring β5-, β5t-, and Cdh24-encoding genes were conserved in the genomes of various

mammalian species, including human, chimpanzee, dog, rat, bat, elephant and horse (NCBI public database). Site #13 was well conserved among those species, whereas site #8 appeared less conserved (Supplementary Table 1).

### Foxn1 can bind to a site proximal to β5t-coding sequence.
We examined whether Foxn1 could actually bind to the candidate sites. To do so, we co-transfected HEK293T cells with a plasmid that expressed Foxn1 and a plasmid that contained a mouse genomic region proximal to β5t-encoding gene (Fig. 1c,d). Foxn1 protein was immunoprecipitated from the lysates of transfected cells, and co-precipitated DNA fragments were PCR-amplified for those candidate sites. We detected Foxn1-dependent immunoprecipitation and PCR amplification for site #13 and not the other sites including site #8 (Fig. 1d). The co-precipitation reflected specific binding to the *β5t*-proximal 3-kb genomic sequence because no signals were detected when the transfected plasmid did not contain this 3-kb fragment (Fig. 1e,f). A point mutation in the core sequence of site #13, which destroyed the capability of binding to Foxn1 protein[21], severely diminished the Foxn1-immunoprecipitated signals, whereas an equivalent mutation in the core sequence of site #8 did not affect the signals (Fig. 1e,f). These results indicate that Foxn1 protein can specifically bind to site #13 that is proximally located 80-bp upstream of *β5t* transcription initiation site.

### Foxn1 can enhance transcription via β5t-proximal cis-element.
We next examined whether Foxn1 binding to site #13 might indeed affect the transcription of proximal β5t-encoding gene. Towards this goal, we co-transfected HEK293T cells with a plasmid that co-expressed Foxn1 protein and tdTomato red fluorescence protein and a plasmid that contained the EGFP green fluorescence protein reporter sequence attached to the herpes simplex virus thymidine kinase gene promoter (HSV-tk) and a variety of the mouse genomic sequence 5′ to the β5t-encoding exon (Fig. 2a). In this reporter assay, the HSV-tk promoter was a weak promoter relative to the cytomegalovirus immediate early gene promoter (CMV-EGFP)[22] (Supplementary Fig. 1) and thus was useful for the sensitive detection of transcriptional regulation mediated by a transfected molecule, such as Foxn1. Upon the plasmid transfection, we measured EGFP fluorescence reporter signals in tdTomato-expressing cells that did or did not co-express transduced Foxn1, in order to detect transcriptional regulation mediated by Foxn1 and cis-regulatory elements. We detected a Foxn1-dependent elevation of HSV-tk-driven EGFP reporter expression when the reporter plasmid contained the 3-kb genomic fragment (Fig. 2b and Supplementary Fig. 2a). The Foxn1-dependent elevation of the reporter expression was markedly abrogated by introducing a point mutation in the core sequence of site #13, which destroyed the Foxn1-binding capability (Fig. 1f), but not by introducing an equivalent mutation in site #8 (Fig. 2b). Foxn1 could elevate the HSV-tk-driven EGFP reporter expression when the reporter plasmid contained the 1-kb mouse genomic fragment, which contained site #13, but could not do so when the reporter plasmid contained no genomic fragment (Fig. 2b). A point mutation in the core sequence of site #13 in the 1-kb genomic fragment was sufficient to abrogate the Foxn1-dependent elevation of the reporter expression (Fig. 2b). The mutant Foxn1 protein that lacked a DNA-binding domain[21] (Fig. 2c) failed to elevate the reporter expression (Fig. 2d), reconfirming that the specific binding of Foxn1 protein to site #13 is responsible for the elevation of the reporter expression. These results indicate that Foxn1 binding to site #13 can elevate the transcription of a proximal reporter gene.

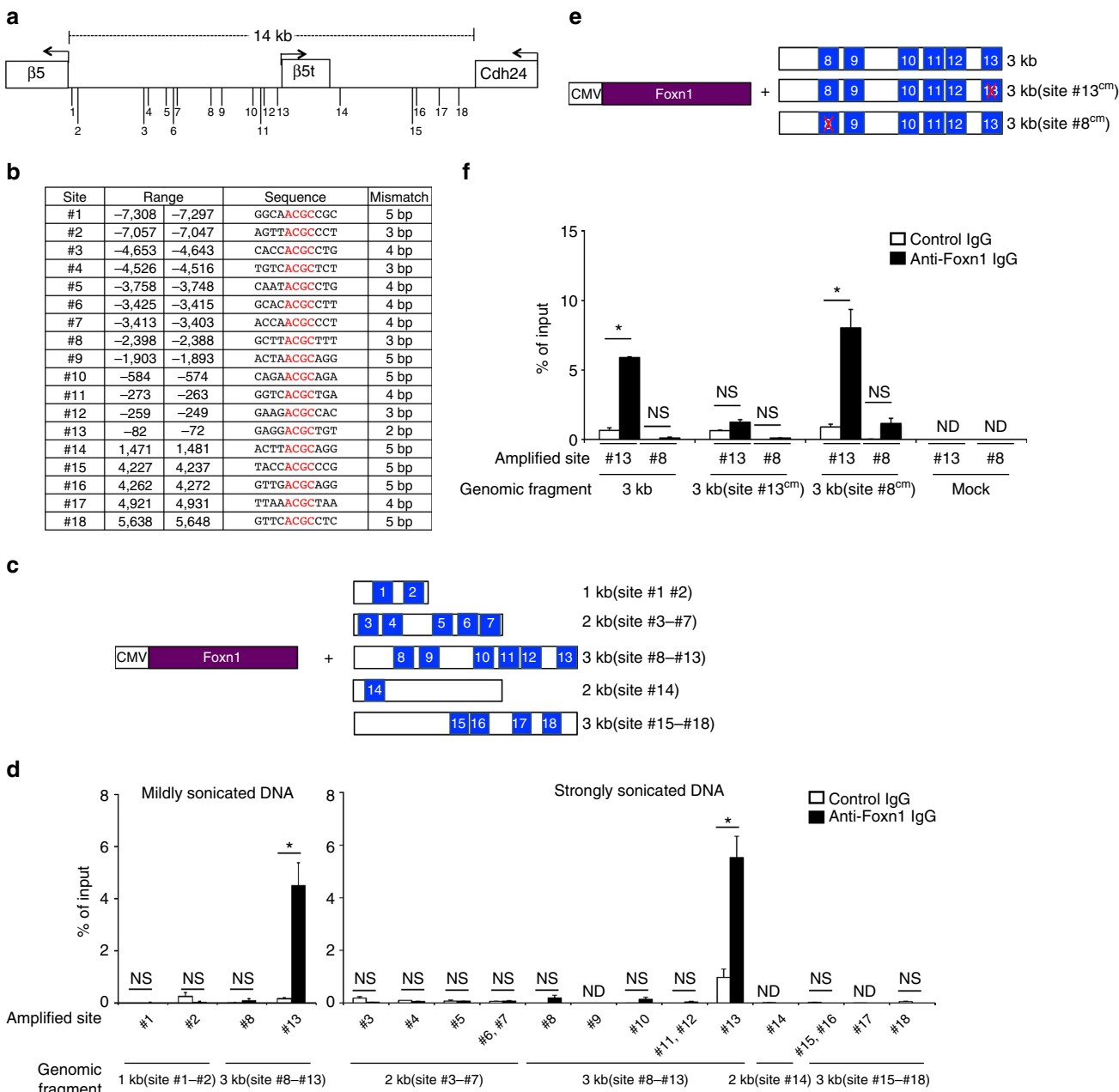

**Figure 1 | Foxn1 binds to β5t-proximal site #13.** (**a**) Schematic diagram of the locations of 18 sites that contain the Foxn1-binding invariant core ACGC tetranucleotide within the 14-kb region proximal to β5t-encoding gene between two neighbouring genes in the mouse genome. Arrows indicate the orientation of the transcription. (**b**) Distances of the 18 sites from β5t translation initiation site are listed. The nucleotide sequences of those sites and their mismatches from the Foxn1-binding consensus 11-bp sequence previously reported[21] are also listed. (**c,d**) HEK293T cells were transfected with a vector that expressed Foxn1 and a plasmid that contained mouse genomic DNA fragment proximal to β5t-encoding gene, as illustrated schematically (**c**). Forty-eight hours after the transfection, formaldehyde-fixed cell lysates that contained protein–DNA complexes were immunoprecipitated with either goat anti-Foxn1 antibody (filled bars) or control IgG (open bars) and PCR-amplified for the indicated candidate sites of the Foxn1-binding sequences. Graphs show the frequency of immunoprecipitated DNA in input DNA (mean ± s.e.m., n = 5), which was sonicated at mild or strong amplitude (**d**). *P < 0.05; NS, not significant; ND, not detectable. (**e,f**) A plasmid that contained the 3 kb DNA fragment upstream of β5t-encoding gene or its variants mutating at the indicated site was used for immunoprecipitation (**e**). A control plasmid that contained no genomic DNA fragment was used where indicated (mock). Graphs show the frequency of immunoprecipitated DNA in input DNA (mean ± s.e.m., n = 3) (**f**). *P < 0.05; NS, not significant; ND, not detectable. All statistical analyses were performed by student's t-test.

To further characterize Foxn1-dependent cis-regulatory elements, we generated a series of luciferase reporter constructs that contained the mouse β5t 5′ genomic region and measured luciferase activity of cells expressing each construct in response to Foxn1 expression. Foxn1-mediated transcriptional activity was readily detectable with constructs that contained 5′-upstream to nucleotide position from −1 to −503, and its magnitude was comparable to that of construct containing longer sequences (Fig. 2e), suggesting that the 503-bp β5t-upstream sequence containing sites #11, #12 and #13 is sufficient for the

Foxn1-dependent promoter activity. We then deleted these three sites from the 503-bp region and examined luciferase activity. We found that the deletion of site #13 most profoundly decreased the Foxn1-dependent promoter activity (Fig. 2f). The deletion of site #12 less severely affected the activity, whereas the deletion of site #11 showed no significant effect on the promoter activity (Fig. 2f). These results reconfirm that site #13 is a potent cis-regulatory

element for Foxn1-mediated reporter transcription, and further suggest the additional roles of other sites, including site #12.

**Induced mutation in proximal Foxn1-binding site in mouse.** Our results suggested the possibility that the binding of Foxn1 to the #13 cis-regulatory site would contribute to the expression of

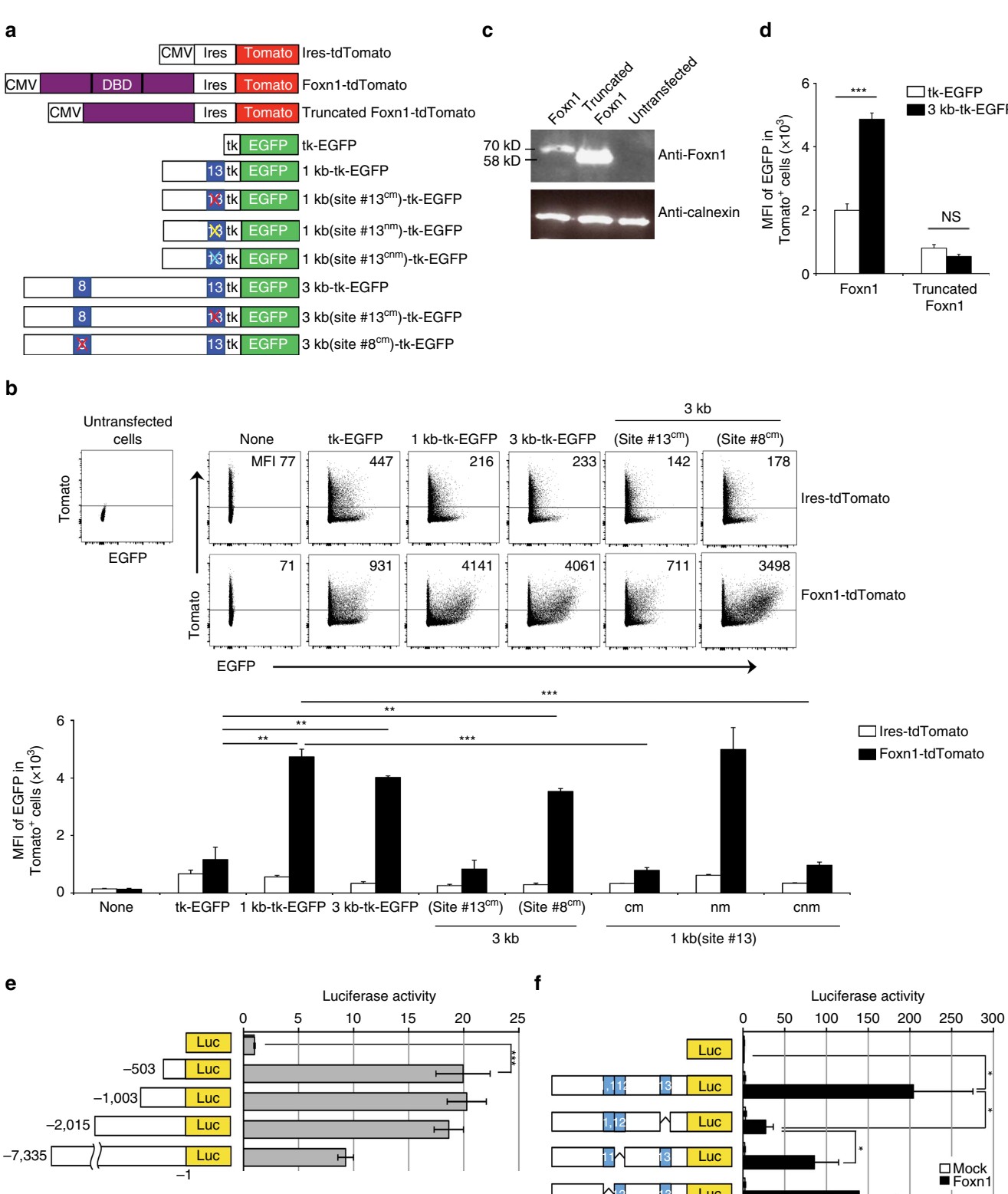

β5t in cTECs. Indeed, *in vivo* chromatin immunoprecipitation analysis showed Foxn1 binding to site #13 in the thymus but not the liver from fetal mice (Fig. 3a). Foxn1 binding to site #13 was readily detectable in isolated cTECs but not isolated medullary TECs (mTECs) (Fig. 3b). That Foxn1 binding to site #8 and site #18 was almost undetectable demonstrated the specificity of *in vivo* Foxn1 binding to site #13 (Fig. 3a,b). The detection of *in vivo* Foxn1 binding to site #13 specifically in cTECs but not mTECs, in which Foxn1 is expressed[23,24], suggests the contribution of epigenetic mechanisms that allow *in vivo* Foxn1 binding to site #13 in cTECs but limit it in mTECs. In this regard, it is interesting to note that the expression levels of *Foxn1* and *β5t* in freshly isolated cTECs are well correlated under different cell culture conditions (Fig. 3c). When cTECs from the thymus lobes were dispersed, *β5t* messenger RNA (mRNA) expression in cTECs was quickly lost within 12 h (Supplementary Fig. 3a). *β5t* mRNA expression remained low in two-dimensional (2D) flat dish culture for up to 72 h but partially recovered in reaggregation thymus organ culture (RTOC) after 48 h (Supplementary Fig. 3a). Microarray analysis of embryonic thymic stromal cells cultured in either 2D flat dish culture or RTOC for 72 h revealed that three transcription factors, *Foxn1*, *Hey1* and *Spatial*, behaved similarly to *β5t* in these culture conditions (Fig. 3c, Supplementary Fig. 3b), which was confirmed by the quantitative mRNA analysis (Supplementary Fig. 3c). Among these three transcription factors, *Foxn1* but not *Hey1* or *Spatial* promoted the reporter expression in HEK293T cells through the *β5t* 5′ genomic region (Fig. 3d). These results reinforce the possibility that Foxn1 directly activates *β5t* expression in cTECs.

To directly examine whether Foxn1 binding to site #13 is functionally relevant *in vivo*, we introduced a point mutation into the site #13 sequence in the mouse genome and examined the phenotypes of those mutant mice. An improved CRISPR/Cas9-mediated genome editing technology[25] was used to introduce the point mutation in the mouse genome. Three independent alleles of mutant mouse strains generated in this study identically contained the intended point mutation in site #13 (Fig. 3e), at the functionally relevant nucleotide in Foxn1-binding and Foxn1-dependent reporter transcription (Figs 1f and 2b). Chromatin immunoprecipitation analysis of cTECs isolated from site #13 homozygous mutant mice showed that Foxn1 binding to site #13 in cTECs *in vivo* was significantly ($P < 0.01$; Student's *t*-test) reduced by the introduced mutation in the genome (Fig. 3f). The non-specific cleavage of the off-target sequence is a possible risk of the CRISPR/Cas9-mediated genome editing. The cleavage efficiency is dependent on the number, position and distribution of mismatches[26]. We examined fifteen off-target genomic sequences that exhibited the highest homology to the RNA-guide sequence and the highest scores of the off-target likeliness, and found that all of them remained intact without mutations in all of the three mutant alleles (Supplementary Fig. 4), suggesting that the genome editing carried out in this study introduced no apparent off-target mutations in the mouse genome. The following phenotypes of the mutant mice reported in this study were reproduced in all the three independent alleles.

**Diminished β5t expression in cTECs in mutant mice.** Mice carrying either heterozygous or homozygous alleles for the site #13 mutation were born and fertile. No apparent abnormality in macroscopic appearance was noted. The thymus contained unaffected corticomedullary architectures and their weights were normal (Fig. 4a). Immunofluorescence analysis of the thymic sections and flow cytometric analysis of liberase-digested thymic cells showed that the cellularity of cTECs and mTECs in those mutant animals was undisturbed (Fig. 4b). Importantly, however, cTECs from mice carrying the homozygous mutation at site #13 had markedly reduced β5t expression (Fig. 4c). Flow cytometric analysis of isolated cTECs showed that the β5t expression was reduced significantly ($P < 0.001$; Student's *t*-test) and appeared homogeneous in the homozygous cTECs, whereas the β5t expression was reduced but still significantly ($P < 0.001$; Student's *t*-test) detectable when compared with the background signals detected in β5t-deficient cTECs (Fig. 5a). The expression levels of MHC class I and class II molecules in cTECs and mTECs were not reduced in those mutant mice (Fig. 5b,c). In contrast, β5t expression in cTECs of heterozygous mutant mice appeared unaffected (Fig. 5a). The amount of *β5t* mRNA detectable in isolated cTECs was well correlated with the amount of β5t proteins (Fig. 5d), suggesting that the reduction in β5t expression is due to reduced *β5t* transcription caused by the mutation in site #13. These results indicate that homozygous mutation in site #13 markedly diminishes β5t expression in cTECs *in vivo*.

**Defective CD8[+] T-cell generation in mutant mice.** We finally examined whether and how the site #13 mutation would affect T-cell development in the thymus. The numbers of total thymocytes and CD4[−]CD8[−]TCR[low], CD4[+]CD8[+]TCR[low] and CD4[+]CD8[−]TCR[high] thymocytes were not affected in heterozygous and homozygous site #13 mutant mice (Fig. 6a). However, the cellularity of CD4[−]CD8[+]TCR[high] thymocytes was significantly ($P < 0.05$; Student's *t*-test) reduced in homozygous but not heterozygous mutant mice (Fig. 6a). The reduction in the number of CD4[−]CD8[+]TCR[high] thymocytes in homozygous mutants was significant but less prominent than that in β5t-deficient mice (Fig. 6a). An essentially similar reduction in the number of CD4[−]CD8[+] T cells was observed in the spleen of

**Figure 2 | Foxn1 binding to site #13 enhances transcription of proximal gene.** (**a**) HEK293T cells were transfected with a plasmid that co-expressed Foxn1 protein and tdTomato red fluorescence protein and a plasmid that contained the EGFP green fluorescence protein reporter sequence attached to the herpes simplex virus thymidine kinase gene promoter (HSV-tk) and a variety of the mouse genomic sequence 5′ to β5t-encoding gene as indicated. Site #13 cm, a point mutation in the core sequence of site #13; site #13 nm, a point mutation in the non-core sequence of site #13; site #13 cnm, a mutation in both core and non-core sequence of site #13. (**b**) Dot plots show the expression of Tomato and EGFP in propidium iodide (PI)-negative viable HEK293T cells co-transfected with indicated EGFP reporter vectors and ires-tdTomato-expressing plasmid (upper profiles) or Foxn1-ires-tdTomato plasmid (lower profiles). Numbers in dot plots indicate mean fluorescence intensity (MFI) of EGFP expression in tdTomato[+] PI[-] cells. Bar graphs show MFI (means ± s.e.m., $n = 3$) of EGFP expression in Tomato[+] PI[-] cells. **P < 0.01; ***P < 0.001. See also Supplementary Fig. 2a. (**c**) Immunoblot analysis of Foxn1 protein or mutant Foxn1 protein without the DNA-binding domain (DBD). Calnexin was examined as the loading control. (**d**) HEK293T cells were co-transfected with indicated plasmids. EGFP reporter expression was measured as in **b**. ***P < 0.005; n.s., not significant. (**e**) HEK293T cells were transfected with a series of luciferase reporter constructs that contained indicated lengths of the *β5t* 5′ genomic region, together with a Foxn1-encoding plasmid. Histograms represent relative luciferase activity, where the activity without genomic sequences was set as 1. Means ± s.d. ($n = 3$) are shown. ***P < 0.005. (**f**) Luciferase reporter constructs that lacked site #11, #12 or #13 were tested. Means ± s.d. ($n = 3$) are shown. *P < 0.05. All statistical analyses were performed by student's *t*-test.

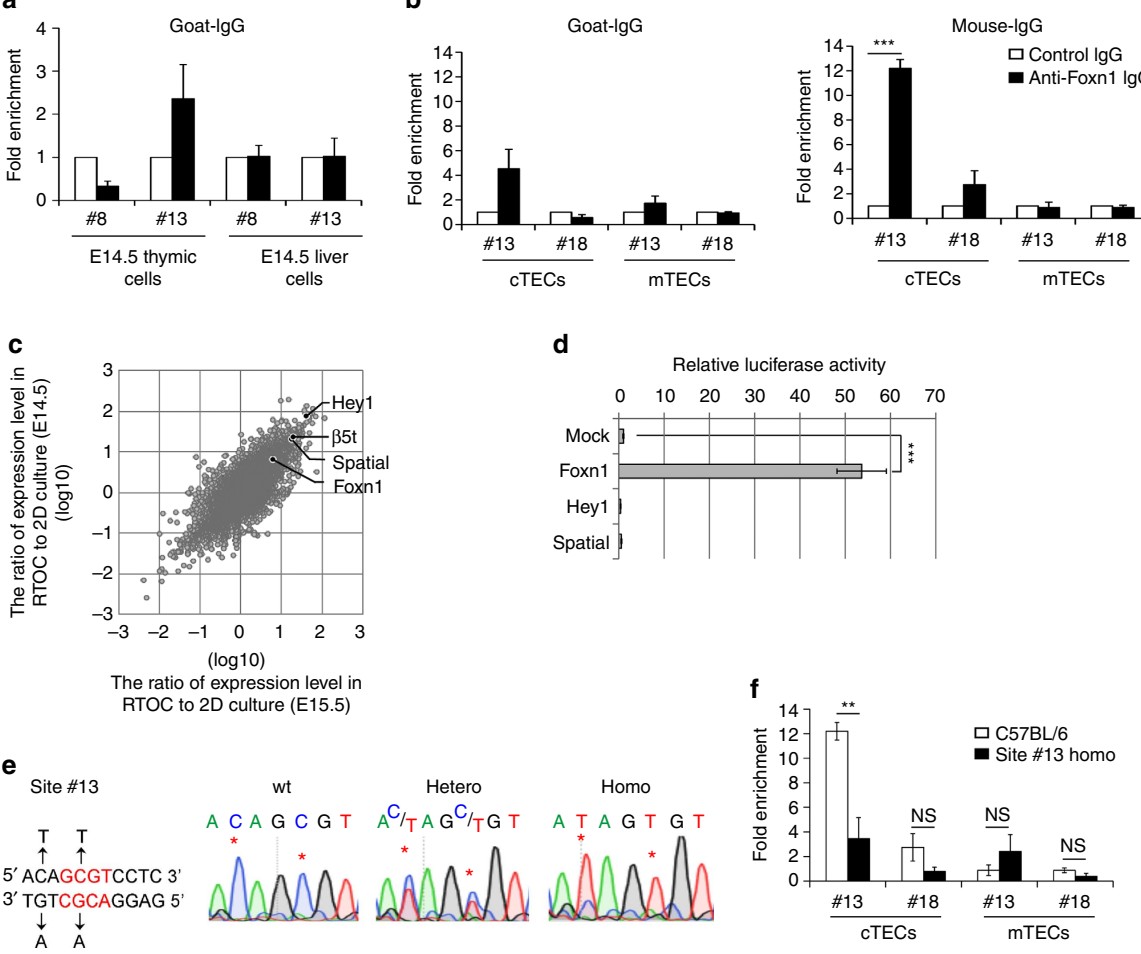

**Figure 3 | Generation of site #13 mutant mice. (a)** Thymuses and livers isolated from E14.5 embryos were liberase-digested. Protein–DNA complexes were immunoprecipitated with goat anti-Foxn1 antibody (filled bars) or control IgG (open bars) and PCR-amplified for site #8 or site #13. Graph shows fold enrichment (means ± s.e.m., $n = 9$) of anti-Foxn1-precipitated signals normalized to the signals by control IgG. **(b)** $CD45^- CD326^+ UEA1^- CD249^+$ cTECs and $CD45^- CD326^+ UEA1^+ CD249^-$ mTECs were isolated from 2-week-old C57BL/6 mice. Protein–DNA complexes were immunoprecipitated with anti-Foxn1 antibody (filled bars) or control IgG (open bars) and PCR-amplified for site #13 or site #18. Graph shows fold enrichment (means ± s.e.m., $n = 3$) of anti-Foxn1-precipitated signals normalized to the signals by control IgG. ***$P < 0.001$. **(c)** Comparison of gene expression profiles between RTOC and 2D culture of thymic stromal cells from E14.5 and E15.5 mice by microarray analysis. Grey dots represent ratios (log scale) of gene expression levels (33,749 transcripts) in the two culture conditions. Longitudinal and horizontal axes show the ratios in E14.5 and E15.5 thymic stromal cells, respectively. **(d)** A plasmid encoding Foxn1, Hey1 or Spatial was transfected into HEK293T cells together with a firefly luciferase that contained the 7-kb β5t 5′-flanking region and a control plasmid encoding Renilla luciferase. Intensities of firefly and Renilla luciferase activities were measured 48 h after transfection. Histograms represent relative luciferase activity, where the activity in mock transfection was set as 1. All data are shown as means ± s.d. ($n = 3$). ***$P < 0.005$. **(e)** Mutant mice carrying the mutation in site #13 proximal to β5t-encoding gene were generated by the CRISPR/Cas9-mediated genome editing technology. The 2-bp mutation at site #13 (left) was confirmed in wild-type (wt), heterozygous, and homozygous mutant mice (right). **(f)** cTECs and mTECs were isolated from 2-week-old site #13 homozygous mutant mice. Graph shows fold enrichment (means ± s.e.m., $n = 3$) of mouse monoclonal anti-Foxn1-precipitated signals normalized to the signals by control IgG (filled bars) and comparison to the value in TECs isolated from C57BL/6 mice (open bars) as shown in **b**. **$P < 0.01$; NS, not significant. All statistical analyses were performed by student's $t$-test.

homozygous mutant mice (Fig. 6b). These results indicate that homozygous mutation in site #13 significantly diminishes the generation of $CD8^+$ T cells in the thymus.

A recent report described that $CD4^+ CD8^+$ thymocytes could be divided into three subpopulations on the basis of TCRβ and CD5 expression levels; $TCRβ^{low} CD5^{low}$ DP1 thymocytes are enriched with pre-selection thymocytes; $TCRβ^{intermediate} CD5^{high}$ DP2 thymocytes are enriched with recently TCR-engaged thymocytes and contain both MHC class I- and MHC class II-restricted thymocytes; and $TCRβ^{high} CD5^{intermediate}$ DP3 thymocytes are enriched with MHC class I-engaged positively selected thymocytes that give rise to $CD4^-$

$CD8^+ TCR^{high}$ thymocytes[27]. We detected a significant ($P < 0.05$; Student's $t$-test) reduction in DP3 thymocytes in homozygous mutant mice compared with heterozygous mutant mice, whereas the reduction in DP3 thymocytes was less severe than that in β5t-deficient mice (Supplementary Fig. 5). These results suggest that the homozygous mutation in site #13 disturbs positive selection of MHC class I-restricted $CD4^- CD8^+$ thymocytes.

## Discussion

The thymoproteasome component β5t has a pivotal role in the generation of functionally competent $CD8^+$ T cells and is

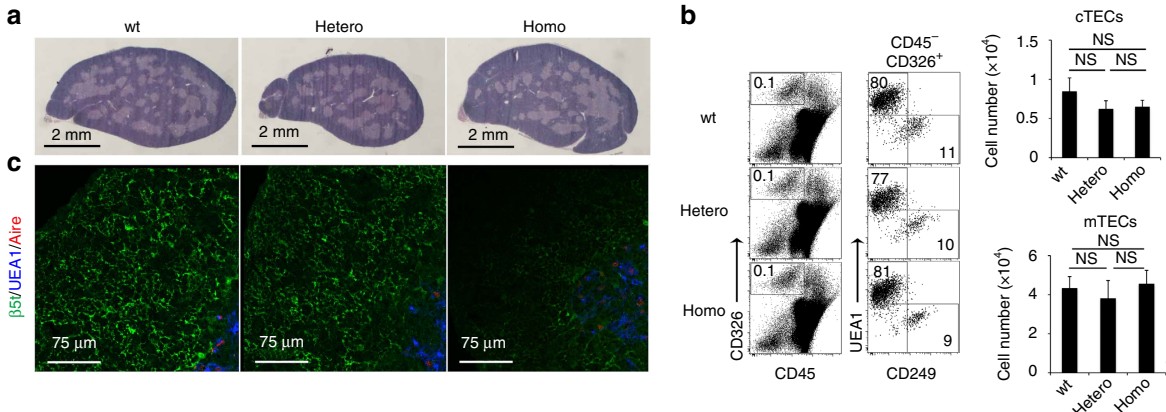

**Figure 4 | Diminished β5t expression in thymus of site #13 mutant mice. (a)** Haematoxylin and eosin staining of thymic sections from 2-week-old mice. Representative data from three independent experiments are shown. Scale bar, 2 mm. **(b)** Flow cytometric analysis of liberase-digested thymic cells isolated from 2-week-old mice. Dot plots show CD326 and CD45 expression in total thymic cells (left), and UEA-1 reactivity and CD249 expression in CD45⁻CD326⁺-gated epithelial cells (middle). Bar graphs show cell number (means ± s.e.m., $n = 4$) of CD45⁻CD326⁺UEA1⁻CD249⁺ cTECs and CD45⁻CD326⁺UEA1⁺CD249⁻ mTECs. NS, not significant. Statistical analyses were performed by student's t-test. **(c)** Immunofluorescence analysis of β5t (green), Aire (red) and UEA-1-binding molecules (blue) in thymic sections from 2-week-old mice. Representative data from three independent experiments are shown. Scale bar, 75 μm.

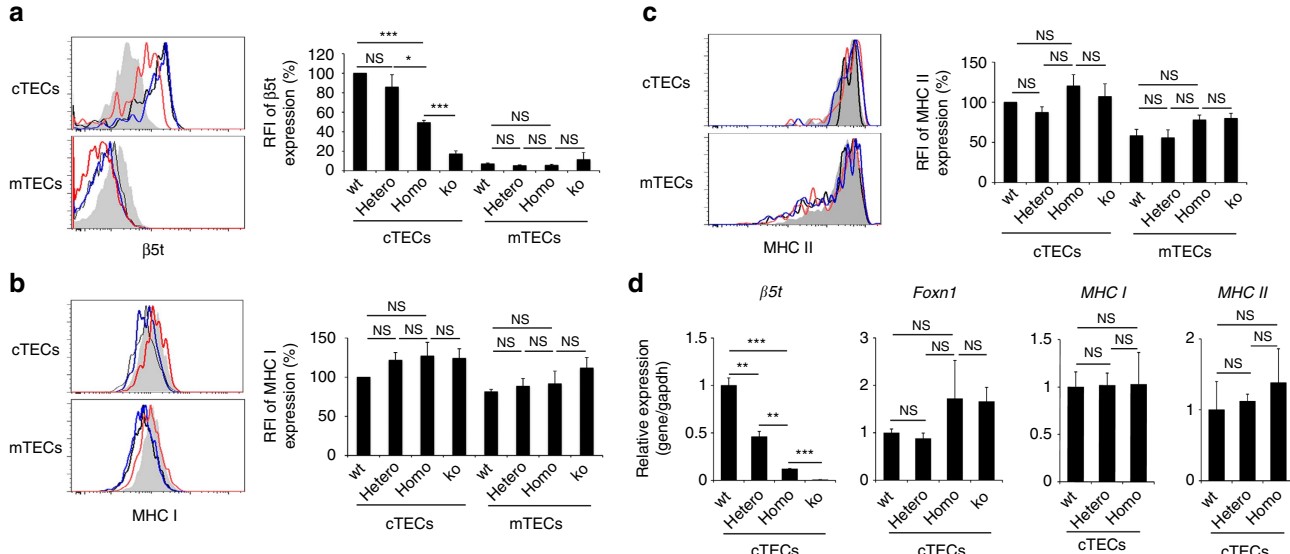

**Figure 5 | Diminished β5t expression in cTECs of site #13 mutant mice. (a–c)** Histograms show the expression of β5t **(a)**, MHC class I **(b)** and MHC class II **(c)** in cTECs and mTECs of wild-type (black lines), heterozygous mutant (blue lines), homozygous mutant (red lines) and β5t-deficient (grey shades) mice at 2 weeks old. Bar graphs show the relative fluorescence intensity (RFI, $n = 4$) of β5t **(a)**, MHC class I **(b)**, and MHC class II **(c)** expression normalized to the mean fluorescence intensity measured in wild-type cells. *$P < 0.05$; ***$P < 0.001$; NS, not significant. See also Supplementary Fig. 2b. **(d)** Relative mRNA levels (means ± s.e.m., $n = 3$) of β5t, Foxn1, MHC I and MHC II in cTECs isolated from 2-week-old mice were measured by quantitative reverse transcription–PCR. mRNA levels were normalized to those of Gapdh mRNA levels and are shown relative to the levels in wild-type cTECs. **$P < 0.01$; ***$P < 0.001$. All statistical analyses were performed by student's t-test.

uniquely expressed in the thymus. However, how the thymus-specific β5t expression is regulated was unknown. The present results show that the mouse genomic sequence termed site #13, located 80-bp upstream of β5t transcription initiation site in chromosome 14, is well conserved in the genomes of various mammalian species and functions as a Fonx1-dependent cis-regulatory element in the transcriptional promotion of β5t gene expression. CRISPR/Cas9-mediated editing of mouse genome reveals that a point mutation in site #13 reduces β5t expression in cTECs and diminishes the cellularity of CD8⁺ T cells generated in vivo. Our chromatin immunoprecipitation analysis reconfirms the in vivo binding of Foxn1 to site

#13 in cTECs and the reduction in the binding by the point mutation in site #13. Altogether, these results reveal a Foxn1-binding cis-regulatory element that has a pivotal role in thymic epithelial cell dependent positive selection of CD8⁺ T cells. It has been suggested that Foxn1 regulates the expression of several genes, including DLL4, CCL25 and PD-L1 (refs 17–20), as well as β5t (refs 11,14), based on the reduced expression of those genes in the embryonic thymus primordium of Foxn1-deficient mice. In contrast, the present results unveil a cis-regulatory element that Foxn1 directly acts on to promote the transcription of a functionally relevant gene in TECs. Our results establish that Foxn1 directly enhances the transcription of

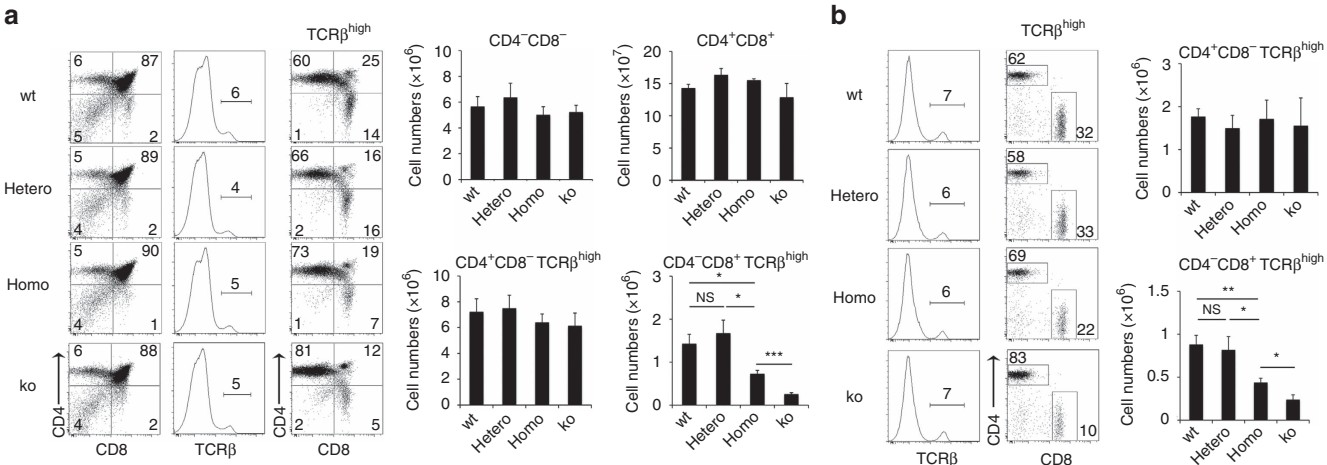

**Figure 6 | Defective CD8$^+$ T cell production in site #13 mutant mice. (a)** Flow cytometric analysis of thymocytes from 2-week-old mice. Shown are dot plots for CD8 and CD4 expression (left) and TCRβ expression (middle) in PI$^-$ viable cells and dot plots for CD8 and CD4 expression in PI$^-$ TCRβ$^{high}$ cells (right). Bar graphs show cell numbers (means ± s.e.m., $n = 4$–6) of indicated thymocyte populations. **(b)** Flow cytometric analysis of splenocytes from 2-week-old mice. Histograms show TCRβ expression in PI$^-$ viable cells. Dot plots show CD8 and CD4 expression in PI$^-$ TCRβ$^{high}$ cells. Bar graphs show numbers (means ± s.e.m., $n = 4$–6) of CD4$^+$CD8$^-$ TCRβ$^{high}$ T cells and CD4$^-$CD8$^+$ TCRβ$^{high}$ T cells. Numbers in dot plots and histograms indicate frequency of cells within indicated area. *$P < 0.05$; **$P < 0.01$; ***$P < 0.001$; NS, not significant. Statistical analyses were performed by student's $t$-test. See also Supplementary Fig. 2c,d.

β5t, thereby directly controlling the thymus-dependent production of CD8$^+$ T cells.

Interestingly, β5t is abundantly expressed in cTECs, but is not detectable in other cells including skin epithelial cells[11–13], in which Foxn1 is expressed and important for hair follicle development[28,29]. It is, therefore, reasonable to assume that Foxn1 does not always induce β5t expression regardless of cellular context. Instead, the expression of β5t may require an unknown cellular context that is unique in cTECs, in addition to the expression of Foxn1. In this regard, it is interesting to note that β5t is not detectable in the majority of mTECs, even though it is abundant in cTECs[11–13]. Unlike β5t, however, Foxn1 is readily detectable in most mTECs and cTECs, and the development of mTECs as well as of cTECs is dependent on Foxn1 (refs 23,24,30,31). The disparity in expression between β5t and Foxn1 in mTECs further supports the possibility that β5t expression is additionally regulated by mechanisms other than the Foxn1-mediated transcriptional promotion. β5t is expressed in bipotent TEC progenitors that give rise to cTECs and mTECs and the vast majority of mTECs are derived from β5t-expressing progenitor cells[12,13,32]. It is, therefore, possible that the additional mechanism regulating β5t expression may involve termination specifically in mTECs and/or maintenance specifically in cTECs. Alternatively, it is also possible that the difference in Foxn1 expression levels in cTECs and mTECs may account for the difference in β5t expression in those cells, because higher expression levels of Foxn1 are detectable in cTECs than mTECs[23,24,33] (our unpublished results). In this context, it is interesting to note that our *in vivo* chromatin immunoprecipitation results demonstrated that Foxn1 binding to site #13 was clearly detectable in cTECs but not mTECs, suggesting that mTEC-specific epigenetic modification of the β5t-encoding genomic region may limit access of Foxn1 to site #13 in mTECs.

A recent report described that Foxn1 binds to many sites in the mouse genome, including the sites proximal to β5t gene, and that Foxn1 is capable of regulating β5t transcription in the *in vitro* reporter assay[34]. The present results unveil a cis-regulatory element that Foxn1 directly acts on to promote β5t transcription in cTECs *in vivo*. We would like to reiterate that the direct target of Foxn1 in controlling the transcription of a functionally relevant gene in TECs *in vivo* has been revealed in this study.

Finally, our results showing that β5t expression in cTECs of site #13 homozygous mutant mice was ~50% of the normal expression levels in cTECs of control mice was in concordance with the reduction in the number of CD4$^-$CD8$^+$TCR$^{high}$ thymocytes to ~50% of normal cellularity in control mice. On the other hand, no significant reduction in β5t expression or CD4$^-$CD8$^+$TCR$^{high}$ thymocyte cellularity was detectable in site #13 heterozygous mutant mice. These results suggest that the cellularity of CD4$^-$CD8$^+$ thymocytes positively selected in the thymus is determined by the availability of β5t-dependent peptide-MHC complexes expressed by cTECs, a possibility that further suggests that the availability or avidity, in addition to the affinity, of peptide-MHC complexes contributes to inducing β5t-dependent positive selection of CD8$^+$ T cells in the thymus. The novel Foxn1-β5t transcriptional axis presented in this study is expected to provide the basis for better understanding and future manipulation of the thymus-dependent generation and regeneration of functionally competent T cells.

## Methods

**Genome sequencing and analysis.** Genome DNA was PCR-amplified and sequenced by using Big Dye Terminator V3.1 cycle sequencing kit (Applied Biosystems), and analysed by Genetic Analyser 3500 (Applied Biosystems). Genome sequences registered in the NCBI public database were analysed using Genetyx and mVista[35,36].

**Constructs and transfection.** Full-length Foxn1 complementary DNA was PCR-amplified from C57BL/6 mouse cTECs by PrimeSTAR DNA polymerase (Takara) and cloned into pCR-blunt vector (Invitrogen) and into CMV-promoter-driven bicistronic ires-tdTomato-containing plasmid. Genomic fragments PCR-amplified from C57BL/6 mouse genomic DNA were cloned into HSV-tk-vector-driven EGFP reporter plasmid. Point mutations in the reporter plasmids were introduced with a PrimeSTAR mutagenesis basal kit (Takara). HEK293T cells were cultured in Dulbecco's modified eagle medium supplied with 10% fetal bovine serum and 100 U ml$^{-1}$ penicillin streptomycin at 37 °C and 5% CO$_2$. Cells were transfected using X-tremeGENE nine DNA transfection reagent (Roche).

**DNA and chromatin immunoprecipitation.** Transfected cells or liberase-digested tissues isolated from E14.5 C57BL/6 mice were fixed in 1% formaldehyde for 10 min, neutralized with 125 mM glycine, and lysed with lysis buffer (1% NP-40, 1% Triton-X, 50 mM Tris-HCl, pH 8.0, 10 mM EDTA) supplemented with the protease inhibitor cocktail (Sigma) for 20 min. Lysates were sonicated at 30% amplitude for five cycles (strong amplitude) or 20% amplitude for three cycles (mild amplitude) of 20 s on and 60 s off (Branson Sonifier). DNA was pre-cleared with 50% protein G-Sepharose (GE Healthcare) in salmon sperm DNA (Sigma). DNA–protein complex was immunoprecipitated with 2 µg of goat anti-Foxn1 polyclonal antibody (G-20, Santa Cruz) or control goat polyclonal IgG (Abcam), heated at 65 °C for 4 h, and treated with proteinase and RNase. Immunopreciptated DNA was ethanol-extracted and quantitated by quantitative PCR.

**Chromatin immunoprecipitation in thymic epithelial cells.** Formaldehyde-fixed cTECs and mTECs isolated from 2-week-old mice were lysed with RIPA buffer (0.1% SDS, 1% Triton X-100, 0.1% Na-DOC, 10 mM Tris-HCl, pH8.0, 1 mM EDTA, 140 mM NaCl) containing protease inhibitor cocktail. Lysates were sonicated in a Covaris S220 (Covaris). Immunoprecipitation was performed as previously described[37]. Briefly, DNA–protein complex was immunoprecipitated with 2 µg of mouse anti-Foxn1 monoclonal antibody[24], or control mouse IgG, coupled to Dynabeads protein G (Veritas). DNA–protein complex was heated at 65 °C for 12 h. Immunoprecipitated DNA was purified with a Qiaquick PCR Purification Kit (Qiagen) and quantitated by quantitative nested PCR.

**Culture of thymic stromal cells.** To obtain thymic stromal cells, thymic lobes from E14.5 or E15.5 C57BL/6 mouse fetuses were cultured in the presence of 1.35 mM deoxyguanosine for 7 to 9 days and then enzymatically dispersed with 0.125% trypsin for 30 min at 37 °C, as previously described[38]. For RTOC, $10^6$ thymic stromal cells were resuspended in 10 µl of culture medium (RPMI1640, 10% fetal bovine serum, 100 U ml$^{-1}$ penicillin, and 100 µg ml$^{-1}$ streptomycin) and placed on Nuclepore Track-Etch Membrane (Whatman). For 2D culture, $10^6$ thymic stromal cells were plated onto a 35-mm culture dish. Cell cultures were incubated at 37 °C in 5% $CO_2$.

**In vitro transcription reporter assay.** Forty-eight hours after the transfection, cells were analysed for the expression of fluorescence proteins using FACSVerse (BD Biosciences). For luciferase reporter assay, transcriptional activity was measured using a dual luciferase reporter system (Promega). Genomic fragments were subcloned into pGL4.20 firefly luciferase vector and co-transfected into HEK293T cells along with pGL4.74 Renilla luciferase vector and a plasmid encoding a transcription factor using PEI MAX (Polysciences). Luciferase activity was measured according to the manufacturer's instructions (Promega).

**Immunoblot analysis.** Cell lysates in lysis buffer (1% NP-40, 1% Triton-X, 50 mM Tris-HCl, pH 8.0, 10 mM EDTA) supplemented with the protease inhibitor cocktail (Sigma) were subjected to SDS–polyacrylamide gel electrophoresis, transferred onto the polyvinylidene difluoride membranes (Millipore), and probed with either goat anti-Foxn1 antibody or rabbit anti-Calnexin antibody (Santa Cruz) and horseradish peroxidase conjugated secondary antibody. Signals were detected with ECL reagent (GE Healthcare) and detected with Image analysis system (Atto).

**Microarray analysis.** Fetal thymic stromal cells were cultured in RTOC or 2D culture for 72 h. Total RNA was extracted using RNeasy Mini Kit (QIAGEN). Amplified complementary RNA was labelled using a Low Input QuickAmp Labelling Kit according to the manufacturer's protocol (Agilent Technologies), hybridized to a Whole Mouse Genome Microarray Kit (4 × 44 K; Agilent Technologies, AMADID 014868), washed, and scanned using a SureScan Micro-array scanner (Agilent Technologies). Microarray data were analysed with Feature Extraction software (Agilent Technologies) and then imported into GeneSpring GX software (Agilent Technologies). Probes were normalized by quantile normalization among all microarray data.

**Mice.** β5t-deficient mice were described previously[5]. All mouse experiments were performed with consent from the Animal Experimentation Committee of the University of Tokushima (#13116). Mutant mice were generated as previously described[25]. Briefly, zygotes from (C57BL/6xDBA/2)F1 mice were electroporated with 400 ng µl$^{-1}$ Cas9 mRNA, 200 ng µl$^{-1}$ sgRNA-β5t and 400 ng µl$^{-1}$ single-strand oligodeoxynucleotide for base substitution (ssODN). Electroporated zygotes were transferred into the oviduct of pseudopregnant female mice, and the mutant mice were born on E19. Cas9 mRNA was synthesized using SalI-linearized pSP64TL-hCas9 and an in vitro RNA transcription kit (mMESSAGE mMACHINE SP6 Transcription Kit, Ambion), according to the manufacturer's instructions. A pair of oligo-DNAs targeting β5t (5′-AAACGCTTCTCCACAGCGTCCTCC-3′ and 5′-TAGGGGAGGACGCTGTGGAGAAGC-3′) was annealed and inserted into the BsaI site of pDR274 (Addgene) to produce pDR275-β5t. sgRNA-β5t was synthesized using the DraI-linearized pDR275-β5t as template and the

MEGAshortscript T7 Transcription Kit (Ambion). Synthesized Cas9 mRNA and sgRNA were purified by phenol-chloroform-isoamyl alcohol extraction and isopropanol precipitation. The precipitated RNA was dissolved in Opti-MEM I (Life Technologies) at 2–4 µg µl$^{-1}$, and stored at −20 °C until use. ssODN (5′-CATTTGAGGCCTGGGTCAGCATGGGGAGGAGGAGGAGGACAC TATGGAGAAGCTGGGCTGCAGCCAGAACCAGGGAGTGAG-3′) was purchased from Sigma.

**Thymus section analysis.** Frozen thymuses embedded in OCT compound (Sakura Finetek) were sliced into 5-µm-thick sections. Thymic sections stained with haematoxylin and eosin were examined under a light microscope. For immunofluorescence analysis, the thymuses were fixed in 4% (g/vol) paraformaldehyde and embedded in OCT compound. Frozen thymuses were sliced into 5-µm-thick sections and stained with anti-β5t antibody and UEA-1, followed by AlexaFluor633- and AlexaFluor546-conjugated secondary reagents, respectively. Images were analysed with a TSC SP8 confocal laser-scanning microscope (Leica).

**Flow cytometric analysis and isolation of TECs.** For the analysis of TECs, minced thymuses were digested with 1 unit per ml Liberase (Roche) in the presence of 0.01% DNase I (Roche). Single-cell suspensions were stained for the expression of CD326 (EpCAM, BioLegend), CD45 (eBioscience), CD249 (Ly51, eBioscience), H-2K$^b$ (BioLegend) and I-A$^b$ (BioLegend), and for the reactivity with UEA-1 (Vector Laboratories). For the intracellular staining of β5t, surface-stained cells were fixed in 2% (g/vol) paraformaldehyde, permeabilized in 0.05% saponin, and stained with rabbit anti-β5t antibody followed by AlexaFluor488-conjugated anti-rabbit IgG antibody. For the isolation of TECs, CD45$^-$ cells were enriched with magnetic bead conjugated anti-CD45 antibody (Miltenyi Biotec). For the analysis of thymocytes and splenocytes, cells were stained for the expression of CD4, CD8 and TCRβ (BioLegend). Multicolor flow cytometry and cell sorting were performed on FACSAriaII (BD Biosciences).

**Quantitative reverse transcription–PCR analysis.** Total cellular RNA was reverse-transcribed with oligo-dT primers and SuperScript III reverse transcriptase (Invitrogen). Quantitative real-time PCR was performed using SYBR Premix Ex Taq (TaKaRa) and the StepOnePlus Real-Time PCR System (Applied Biosystems) and LightCycler Probes Master in a LightCycler 480 (Roche). The amplified products were confirmed to be single bands by gel electrophoresis and normalized to the amounts of Gapdh amplification products. The primers used were as follows: β5t, 5′-CTCTGTGGCTGGGACCACTC-3′ and 5′-TCCGCTCTCCCGAA CGTGG-′; Foxn1, 5′-CTCGTCGTTTGTGCCTGAC-3′ and 5′-TGCCTCT TGTAGGGGTGGAAA-3′; MHC I, 5′-CAAGTATACTCACGCCACCC-3′ and 5′-CCCAGTAGACGGTCTTGG-3′; MHC II, 5′-GTACCAGTTCATGGGCG AG-3′ and 5′-CAGGATCTCCGGCTGGCTG-3′; Gapdh, 5′-TTGTCAGCAA TGCATCCTGCAC-3′ and 5′-GAAGGCCATGCCAGTGAGCTTC-3′. For the experiments shown in Supplementary Fig. 3, we used the following primers: β5, 5′-GCTTCACGGAACCACCAC-3′ and 5′-CACCGTCTGGGAAGCAAT-3′; β5t, 5′-CCCAGACCATCCATTCACTT-3′ and 5′-GAAGGTTTGAGGGTCACAG C-3′; Foxn1, 5′-TGGTGCAATAAACTCCCTTACC-3′ and 5′-GGCTTGACCT TGACCTCTGA-3′; Hey1, 5′-CCATCGAGGTGGAAAAGGA-3′ and 5′-CTTC TCGATGATGCCTCTCC-3′; Spatial, 5′-AGCGAGTGACTCATATCCAAGTT-3′ and 5′-GAGCTGGAAAGAGGTGGTGA-3′; G6PD, 5′-GAAAGCAGAGT GAGCCCTTC-3′ and 5′-CATAGGAATTACGGGCAAAGA-3′.

**Statistical analysis.** Statistical comparison was performed with the two-tailed Student's t-test using Prism 6 software (GraphPad).

**Data availability.** Microarray data that support the findings of this study have been deposited in GEO with the primary accession code GSE84222. Genomic sequence data of β5t locus referenced in this study are available in NCBI with the accession code NC_000080.6. The data presented in this study are available from the authors upon reasonable request.

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

## Acknowledgements

We thank Dr Hans-Reimer Rodewald for providing the mouse anti-Foxn1 monoclonal antibody. We also thank Drs Kensuke Takada, Kenta Kondo and Mina Kozai for reading the manuscript and Ms Yukiko Yamashita for technical assistance. This study was supported by grants from MEXT-JSPS Kakenhi (24111004, 23249025 and 16H02630 to Y.T., 25860361 and 15K19130 to I.O., 25221102 to S.M. and 26000014 to K.T.). M.M.U. is supported by a MEXT scholarship for international students.

## Author contributions

Y.T., S.M. and K.T. designed and supervised the study. M.M.U., I.O., R.M., T.N., M.S., J.H. and Y.N. performed the experiments. T.T. generated the mutant mice. I.R. contributed the reagents. I.O., S.M. and Y.T. wrote the paper.

## Additional information

**Competing financial interests:** The authors declare no competing financial interests.

