## [Peer Review File · Nature Communications]

REVIEWERS' COMMENTS:

Reviewer #1 (Remarks to the Author):

My concern in the initial review concerned the limited novel insight into a biological process that is provided by this carefully performed study rather than any shortcomings in the experimentation or the interpretation of results. The authors now provide a further improved version of the MS that should be well suited for publication in Nat Communications.

Reviewer #2 (Remarks to the Author):

The revised manuscript by Uddin et al. incorporates some improvement. The use of a better-characterized anti-Foxn1 antibody certainly strengthens the argument that Foxn1 can bind to the so-called #13 site in the vicinity of the b5t gene. The authors have also excluded some possible off-target effects of their CRISPR/cas9-mediated mutation. The authors also now report the analysis of Foxn1-binding in sorted cTECs and mTECs: the site is not occupied in mTECs, whereas occupancy is readily detectable in cTECs. To explain this, the authors invoke epigenetic mechanisms, which is a plausible idea, but requires further study. It is puzzling that the binding of Foxn1 to the mutated site #13 not completely abolished in cTECs, yet increased in mTECs? Couldn't this be a sign of a more complex factor binding complex at this site? And, if this level of binding is relevant (after all, b5t expression is not fully extinguished in the mutant), then perhaps b5t becomes ectopically expressed in mTECs after mutation: Has this been checked? It is a pity that the response of b5t expression after conditional Foxn1 ablation in an established TEC compartment must await further studies, as this would have been a central piece of supportive evidence for the authors' conclusions.

Point-by-point response to the additional questions raised by the reviewer 2

1) It is puzzling that the binding of Foxn1 to the mutated site #13 not completely abolished in cTECs, yet increased in mTECs? Couldn't this be a sign of a more complex factor binding complex at this site?

The binding of Foxn1 to the mutated site #13 was abolished to the level of control values, which were not significantly higher than the value of no binding (fold enrichment = 1, Fig. 3f). Also, the increase in mTECs that the reviewer noticed was not significantly higher than the value of no binding (Fig. 3f). To avoid confusions, we added the note of statistically “not significant (n. s.)” to the relevant comparisons in Fig. 3f.

2) And, if this level of binding is relevant (after all, b5t expression is not fully extinguished in the mutant), then perhaps b5t becomes ectopically expressed in mTECs after mutation: Has this been checked?

As described above, our results do not show that “b5t expression is not fully extinguished in the mutant”. Instead, our results show that the binding of Foxn1 to the mutated site #13 was abolished to the level of background values, which were not significantly higher than the value of no binding (Fig. 3f). To avoid the confusion, we added the note of statistically “not significant (n. s.)” to the relevant comparisons in Fig. 3f. Furthermore, despite the reviewer's suggestion that b5t may become “ectopically expressed in mTECs after mutation”, our results show that beta5t is not ectopically expressed in mTECs of our mutant mice (Fig. 5a). To specifically answer the reviewer's question, yes, we checked the possibility and described that it was not the case.